# Prevalence of dural puncture headache after caesarean section at a tertiary hospital in the Gambia

Matthew Anyanwu[1,2]*, Admire Coker[1,2], Simon Donkor[3]

1 Edward Francis Small Teaching Hospital, Banjul, The Gambia, 2 School of Medicine and Allied Health Sciences University of The Gambia Kanifing, The Gambia, 3 Medical Research Laboratories, Fajara, The Gambia

* anyanwum@yahoo.com, manyanwu@utg.edu.gm

## Abstract

### Background

Following a caesarean section performed under spinal anaesthesia, a common complication is post-dural puncture headache (PDPH). Spinal anaesthesia has become the most common anaesthetic procedure during caesarean section in our practice. Therefore, knowing the prevalence and risk factors of PDPH will inform practice and add value in our obstetrics practice.

### Methodology

A cross-sectional study was conducted at Edward Francis Small Teaching Hospital Banjul and data was effectively collected from August to October 2023. A structured data collection tool was used. The data was entered into a computer database and analyzed using the SPSS version 26. Pain was graded using a 10-cm visual analogue scale. Results were expressed in simple descriptive statistics and test of significance was set at p-value 0.05.

### Results

A total of 89 participants with mean age of 28 years (SD±6.1) and majority 45(50.6%), between 21 to 30 years. Majority had low parity (0 to 3), 64(71.9%), emergency CS of 71(79.8%), while only 20.2% (n=18) were elective CS. Overall prevalence of Post Dural Puncture Headache (PDPH) was 42.7% with majority of cases presenting with Occipital headache (71%), lasting for 3hrs (42%). A statistically significant association between PDPH and Gestational age with *p-value (p=0.02)*; number of attempts with *p-value (p=0.01);* larger needle gauge and number of CSF drops (p=0.01), respectively were observed.

**Data availability statement:** All relevant data are within the paper and its Supporting Information files.

**Funding:** The author(s) received no specific funding for this work.

**Competing interests:** The authors have declared that no competing interests exist.

## Conclusion

Prevalence of PDPH was high and associated with the use of large needle gauges, multiple attempts and increased CSF drops. Patients that underwent emergency CS, have a higher risk of developing Post-Dural Puncture Headache.

## Introduction

Post-Dural puncture headache (PDPH) is a headache that may occur 24–48 hours after spinal anaesthesia (or inadvertent epidural subarachnoid puncture). Spinal anaesthesia is a neuraxial anaesthesia technique in which local anaesthetic is placed directly in the intrathecal space (subarachnoid space). The subarachnoid space houses sterile cerebrospinal fluid (CSF), the clear fluid that bathes the brain and spinal cord [1]. The practice of using subarachnoid space block (spinal anaesthesia) has become the main stay of anesthetic method in most caesarean sections in our setting [2]. Patient's satisfaction of being awake during surgery is well documented but the practice is not devoid of complications of which one of such complications is PDPH [3]. The pathophysiology of PDPH has been linked to continuous leak of CSF through the spinal puncture hole, decompressing the subarachnoid space with secondary stretching of the cranial nerves, meninges and blood vessels with subsequent vasodilation of the cerebral vasculature [4,5].

The continuous leakage of CSF through the dural puncture hole may produce low CSF pressure and the choroid plexus is unable to secrete sufficient fluid to maintain the CSF pressure. This results to Venous dilation [5] and compensatory increase in brain volume will result in brain sag which in turn will exert traction and stimulate pain sensitive anchoring structures like dural vessels, basal dura and tentorium cerebelli, causing post spinal headache. This condition is found to be more frequent in young females and is said to be associated with needle size and type.

Symptoms include a throbbing headache related to posture and presents most commonly within 48 hours of CSF hypotension which in turn leads to intracranial venous dilation resulting in an increase in brain volume [3].

Regarding prevalence of PDPH, there are many studies in this field. Much more has been published since spinal anaesthesia has become a predominant anaesthetic procedure for caesarean section in our sub-region. Some prospective cohort studies conducted in rural India [5] and Northern Nigeria [6] reported 11.4% and 15.8% incidence of PDPH respectively. However, a retrospective cohort study conducted in Jordan reported a lower incidence of PDPH (6.3%) [7].

A cross sectional study conducted in Ethiopia had incidence of post dural puncture headache 31.3% [8].

Several risk factors have been attributed to PDPH which includes; age, weight, female gender, pregnancy, needle size and design, and number of puncture attempts. Spinal needle size and design has been implicated as risk factors of PDPH [9]. Additionally, insertion of the needle with the bevel parallel to the dural fibers facilitates closure of the hole [10]. Women for caesarean delivery are at increased risk because

of their young age and increased vascular distension response to CSF leakage, due to higher oestrogen level during pregnancy [11]. Multiple dural punctures caused by unsuccessful puncture would increase the rate of PDPH [12].

In view of these risk factors that potentially prevails in our practice and the increasing use of spinal anesthesia for caesarean section, we decided to conduct the study to show prevalence of post dural puncture headache and associated symptoms.

## Methodology

A cross-sectional study design was used to conduct this research. The proposal was submitted for approval in July and recruitment and data collection effectively started on the 13/08/2023 and ended 05/10/2023. Compilation and analysis was completed on the 16/10/2023.

Sample size was calculated with G power 3.1 software. Fig 1; shows graphic representation of the sample size calculation. A sample size of 100 with effect size at 25% = 0.25, Power at 85, t-test; alpha-0.05.

Protocol: Patients were recruited at the labour or antenatal ward when decision to conduct caesarean section was reached. They were followed to theatre and the interest was on spinal needle type, how many attempts and site of insertion of spinal needle. In the postnatal ward they were assessed for headache and pain threshold was graded using a 10-cm visual analogue scale.

### Inclusion and exclusion criteria

Women who were scheduled for either emergency or elective caesarean section with the ages of 18 – 49 years. The women who had pre-existing diagnosis of chronic headaches or migraines were excluded as these conditions may complicate the interpretation of PDPH occurrence and symptoms. Those below 18 years were excluded however no participant is less than 18 that was recruited.

### Data collection

The study was conducted with modified self-administered structured questionnaire which has both discrete and continuous variables. The biodata, reproductive and social history; spinal anaesthetic characteristics and pregnancy outcomes

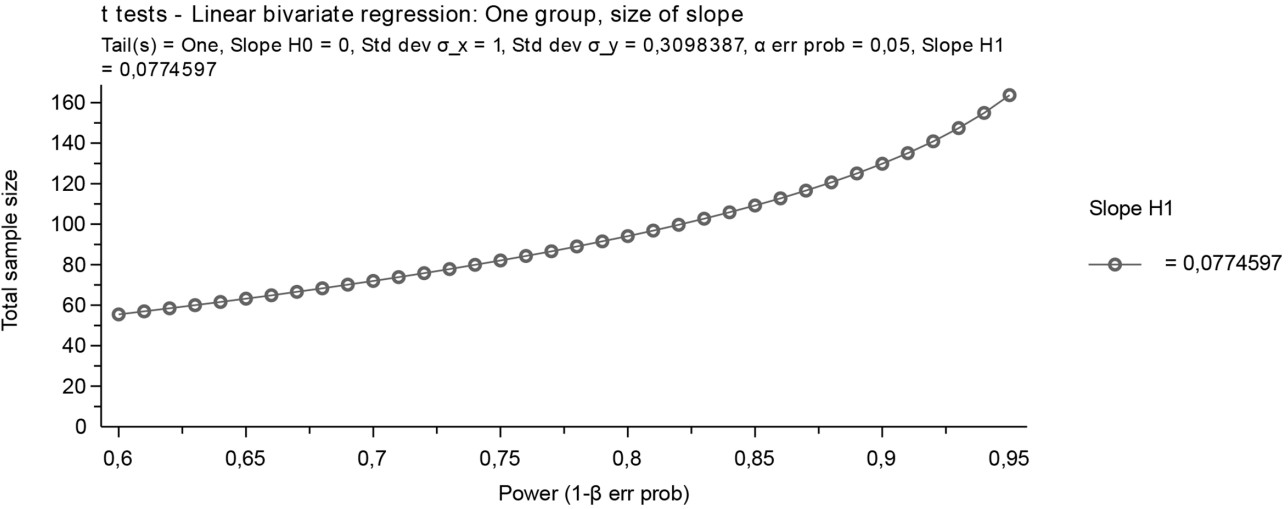

**Fig 1. Graphic representation of the sample size calculation.**

were the variables in this research. The outcome measures includes the sociodemographic characteristics, prevalence of PDPH, the associated symptoms and risk factors of PDPH.

Consent form was used and all patients signed the consent form, no minor was in the study.

## Data analysis

Data was entered into computer database and consistent checks instituted to ensure errors are corrected before analysis. Four [4] data was excluded due to error in data collection and checklist used, so 89 was analyzed. Data was analyzed with SPSS and EPI Info and results expressed by simple descriptive statistics in tables and graphs. The test of significance was represented with p-value at 0.05.

## Ethical considerations

Ethical approval was requested and secured from Research and Ethics Committee of Edward Francis Small Teaching Hospital. The letter of approval was used to have access to patients and medical files at the Obstetrics and Gynaecology department. Informed consent was obtained from all participants and personal information were confidential.

## Results

From 93 patients who were recruited for this study, 4 of them were excluded due to error in data collection and inconsistent checklist that was applied to them. A total of 89 participants data were included for analysis. Table 1 shows, 89 patients, majority belong to the age group between 21–30 years, 45(50.6%), followed by the age group between 31–40 years, 28 (31.5%) and age group between 41–50 years, 2(2.2%). The mean age of 28 years (SD ± 6.1). Majority of patients were married, 79(88.8%), with Muslim religion, 74(83.1%) and a secondary educational level 39(43.8%) predominant.

Table 1. Distribution of women by socio-demographic characteristics. Postnatal ward- EFSTH; August to September 2023. n = 89.

| Socio-demographic characteristics | N (89) | % |
|---|---|---|
| **Age** | | |
| 18 to 20 | 14 | 15.7 |
| 21 to 30 | 45 | 50.6 |
| 31 to 40 | 28 | 31.5 |
| 41 to 50 | 2 | 2.2 |
| **Marital status** | | |
| Married | 79 | 88.8 |
| Single | 10 | 11.2 |
| **Religion** | | |
| Christian | 15 | 16.9 |
| Muslim | 74 | 83.1 |
| **Level of Education** | | |
| Islam/Arabic | 10 | 11.2 |
| Primary | 13 | 14.6 |
| Secondary | 39 | 43.8 |
| Tertiary | 20 | 22.5 |
| No formal education | 7 | 7.9 |

Among 89 women who underwent CS, 38(42.7%) of patients developed PDPH, of which, 16(42.1%) of the headache lasted for >3hrs; 17(44.7%) of them suffered severe pain, and 14(36.8%) of them had moderate pain.

In this study, the most associated symptom was Neck Pain 14 (15.7%) followed by a combination of Back Pain and Neck Pain 9 (10.1%).

There was no statistically significant association with *p-value (p ≤ 0.05)* between Post dural Puncture Headache and the socio-demographic characteristics, except Level of Education with *p-value* of *(p = 0.035)* and *Cramer's V* value of *(v = 0.341)*. Which means a weak association between PDPH and Level of Education.

There is a weak statistically significant association between PDPH and Gestational age with *p-value (p = 0.02)* and *Cramer's V (V = 0.29)* and a weak statistically significant association between PDPH and number of attempts with *p-value (p = 0.01)* and *Cramer's V (V = 0.27)*. There are weak statistically significant associations between PDPH and Level of Puncture and Number of CSF drops, with p-values (p = 0.01) and (p = 0.01) respectively. The Cramer's V are (V = 0.44) and (V = 0.57) respectively. There are no significant associations between PDPH and the rest of the clinical characteristics.

## Discussion

Regarding prevalence of PDPH there are many studies in this field more so now that spinal anaesthesia has become a predominant anaesthetic procedure for caesarean section in our sub-region. In this study the prevalence of PDPH is 42.7% (Fig 2). Some prospective cohort studies conducted in rural India and Northern Nigeria reported 11.4% and 15.8% incidence of PDPH respectively [5,6]. However, a retrospective cohort study conducted in Jordan reported a lower incidence of PDPH (6.3%) [7].

A cross sectional study conducted in Ethiopia had incidence of post dural puncture headache 31.3% and (32.5%) in Cairo University, Egypt [8,13]. These studies were all prospective just like our study, but the population size and the gauge of the needle differs. In this our study the majority (68.5%) had 22G gauge (Table 3) spinal needle which was associated with PDPH with p-value (p = 0.01) in (Table 6). This is similar to what is reported by other scholars working elsewhere [14–16].

However, the smaller the needle size the lower the incidence of PDPH. This was demonstrated by the study conducted in India [5] and Nigeria [6] that used 26G and 25G needles respectively. The sample size also influenced the prevalence reported by different scholars working elsewhere. Our study had small sample size (Fig 1; Table 1) compared to other published reports [17,18] and this may contribute to high prevalence of PDPH we reported.

A retrospective cohort study conducted in Jordan, had a lower incidence of PDPH (6.3%) and repeated attempts increased the risk of its occurrence [7]. The lower incidence could be because it was retrospective and the population size

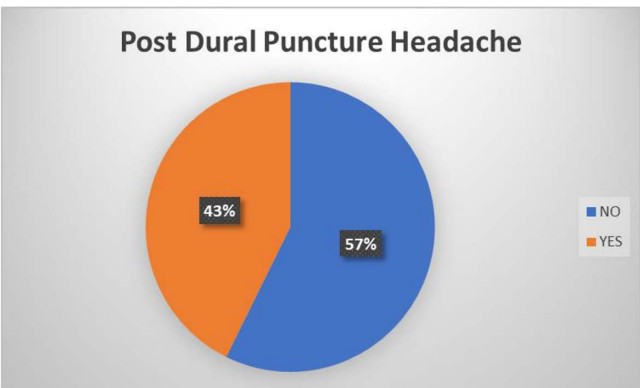

**Fig 2. Prevalence of Post dural Puncture Headache.**

of 630 respondents may have contributed to the lower incidence. Similarly, our study had a slight significant association of PDPH and number of attempts with p-value (p = 0.01) shown in (Table 6).The number of attempts may increase the probability of piercing the dura matter repeatedly, which will increase the volume of CSF leak, thus increasing the probability of development of intracranial hypotension & PDPH. This was a similar findings with other studies [16,19,20] that repeated attempts may increase the risk of developing PDPH. The proportion of repeated attempts of spinal needles related PDPH reports from a population-based study in University of Basel, Switzerland (4.2%) [21] was lower than our prevalence. However, some other studies did not report significant association between number of attempts and PDPH [22,23].

The majority (79.8%) of the caesarean sections (CS) were due to emergency obstetrics conditions which may also have majority of the PDPH (Table 2). The study also revealed that among 89 patients that participated, 18 (20.2%) had elective CS, of which 5 (13.2%) patients developed PDPH, and that 71(79.8%) of patients underwent emergency CS, of which 33 (86.8%) developed PDPH (Table 2). The severity of PDPH reported by patients in this study was higher when compared with other studies conducted in the sub-region. The study conducted in the Northern part of Nigeria [6] and in Ghana [24], most of them had moderate and mild headaches. However, in this study it was observed that the duration of headaches and its severities last for few hours (Table 4), which was not comparable with other studies conducted elsewhere as the headaches last for days (1 to 5 days) in a study conducted in rural India [5].

A recent study shows the incidence of PDPH decreases with higher gauge Quincke needles as follows: 16 to 19 G, about 70%; 20 to 22 G, 20 to 40%; and 24 to 27 G, 5 to 12% [25]. In our study it was predominantly 22G (68.5%) (see Table 3). which has a significant high incidence of PDPH which may explain the high incidence of PDPH we reported. Some studies conducted in Pakistan [26,27] had comparable lower incidence of PDPH because of smaller size spinal needle they used. However, a prospective study conducted in Czech with the use of atraumatic Whitacre and Atre Ucan needles, the incidence of PDPH was remarkably reduced [28]. In this study,

**Table 2. Distribution of women by Clinical characteristics of reproductive health. Postnatal ward-EFSTH; August to September 2023. n = 89Among 89 patients who underwent caesarean section under spinal anesthesia, majority of them had a parity of 0 to 3, 64(71.9%), emergency CS of 71(79.8%), while only 20.2% (n = 18) were elective CS.**

| Clinical characteristics(general). | N (89) | % |
|---|---|---|
| **Parity** | | |
| Para (0 to 3) | 64 | 71.9 |
| Para (4 to 8) | 25 | 28.1 |
| **Previous History of CS** | | |
| No | 67 | 75.3 |
| Yes | 22 | 24.7 |
| **Gestational Age (weeks)** | | |
| 28 to 31 | 2 | 2.2 |
| 32 to 36 | 29 | 32.6 |
| 37 to 41 | 58 | 65.2 |
| **Urgency of CS** | | |
| Elective | 18 | 20.2 |
| Emergency | 71 | 79.8 |
| **Duration in Labour (mins) =71** | | |
| 20 to 35 | 13 | 18.3 |
| 36 to 50 | 46 | 64.8 |
| 51 to 65 | 9 | 12.7 |
| 66 to 80 | 3 | 4.2 |

**Table 3. Distribution of women by anaesthesia procedure.** In this study, 10 mg of Bupivacaine was used in majority 66(74.2%) of the patients and 12.5 mg was used in the minority 23(25.8%). The spinal needle size 22G was used in 61(68.5%) of patients followed by 25G needle used in 28(31.5%) of patients. Two attempts were performed in the majority (69.7%) of the patients. While a third (30.3%) had one attempt. Site of puncture: L3-L4 was the commonest site 74(84.3%). In the majority 56(62.9%) of the patients had 2 drops of cerebrospinal fluid and 1 drop in a third 33(37.1%).

| Clinical characteristics (anaesthesia). | N | % |
|---|---|---|
| **Dosage of Anaesthetic agent (mg)** | | |
| 10 | 66 | 74.2 |
| 12.5 | 23 | 25.8 |
| **Number of attempts** | | |
| 1 | 27 | 30.3 |
| 2 | 62 | 69.7 |
| **Size of needle (G)** | | |
| 22 | 61 | 68.5 |
| 25 | 28 | 31.5 |
| **Level of Puncture** | | |
| L2-L3 | 8 | 9.0 |
| L3-L4 | 75 | 84.3 |
| L4-L5 | 6 | 6.7 |
| **Number of CSF Drops** | | |
| 1 | 33 | 37.1 |
| 2 | 56 | 62.9 |

the most associated symptom was neck pain 14 (15.7%) followed by a combination of back pain and neck pain 9 (10.1%) (Fig 3). Most studies cited in this study assessed immediate impact of spinal anaesthesia on post-dural puncture headache and associated risk factors. But our study also looked at other associated symptom's on those that developed headache.

In Table 5, our study shows no statistically significant association between post dural puncture headache and the socio-demographic characteristics, except some weak association demonstrated with level of education.

In this study, majority of patients experienced severe pain (44.7%) (see Table 4), followed by moderate pain (36.8%), unlike other studies [29,30]. The possible reason could be that this study was done among caesarean section patients which means all were females, that perhaps had low threshold for pain [31].

In our study, the Level of lumbar puncture was mostly done at L3-L4 (84.3%) (Table 6), which had no statistical significant association with PDPH, with p-value (p=0.48). approximately seventeen percent (16.9%) of the study population had previous history of PDPH. There was no significant association between PDPH and previous history of PDPH with a p-value of (p=0.17) see (Table 6).

## Conclusion

This study showed the prevalence and risk factors associated with Post-Dural Puncture Headache (PDPH) following caesarean section. The findings suggests that PDPH is a recognized complication, with a prevalence of approximately 42.7%. It was observed that a higher gestation age and multiple attempts of spinal needle insertion had weak associations with PDPH. The use of larger needle gauges and higher number of CSF drops were identified to have moderate statistical significant relationship with PDPH. Patients that underwent emergency CS, had a higher risk of developing Post-dural Puncture Headache.

**Table 4. Postoperative outcomes.**

| Clinical characteristics (PDPH). | N | % |
|---|---|---|
| **Previous history of PDPH** | | |
| No | 74 | 83.1 |
| Yes | 15 | 16.9 |
| **Severity of previous history** | | |
| Mild | 4 | 4.5 |
| Moderate | 5 | 5.6 |
| Severe | 74 | 83.1 |
| Not sure | 6 | 6.7 |
| **Post Dural Puncture Headache** | | |
| No | 51 | 57.3 |
| Yes | 38 | 42.7 |
| **Severity of PDPH n=(38)** | | |
| Mild | 7 | 18.4 |
| Moderate | 14 | 36.8 |
| Severe | **17** | **44.7** |
| **Duration of Headache n =(38)** | | |
| >3hrs | 16 | 42.1 |
| 1hr | 10 | 7.9 |
| 2hrs | 12 | 31.6 |
| **Location of Headache n =(38)** | | |
| Frontal | 11 | 28.9 |
| Occiput | 27 | 71.1 |

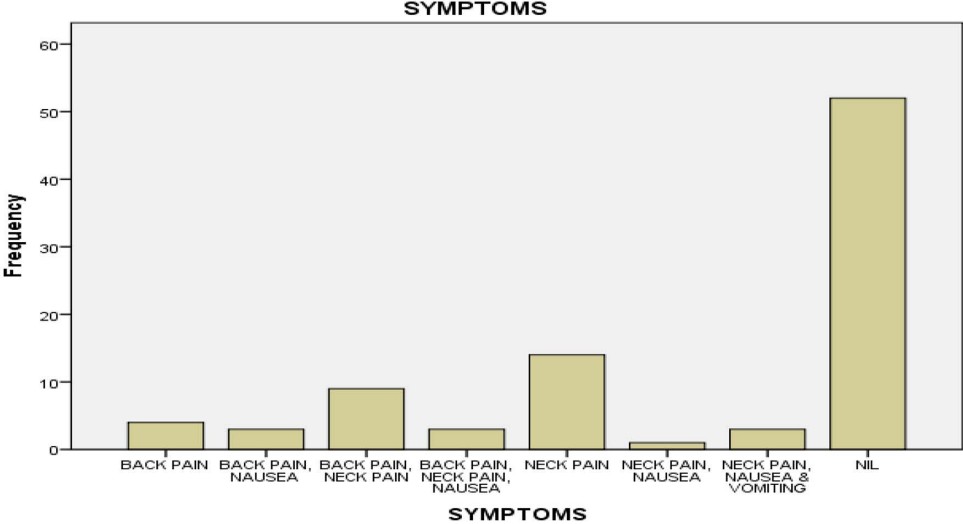

**Fig 3. Associated Symptoms of Post Dural Puncture Headache.**

**Table 5. Relationship between Post Dural Puncture Headache and pregnant women Socio-demographic characteristics. *P<0.05.***

| Socio-demographic characteristics | Post Dural Puncture Headache | |
|---|---|---|
| | *P-Value* | *V(Cramer's)* |
| **Age** | 0.25 | 0.21 |
| **Address** | 0.41 | 0.49 |
| **Marital Status** | 0.38 | 0.09 |
| **Religion** | 0.82 | 0.03 |
| **Level of Education** | 0.035 | 0.34 |

**Table 6. Relationship between Post Dural Puncture Headache and pregnant women clinical characteristics. *P<0.05.***

| Intraoperative clinical characteristics | Post Dural Puncture Headache | |
|---|---|---|
| | *P-value* | *V* |
| Parity | 0.20 | 0.14 |
| Gestational Age | 0.02 | 0.29 |
| Urgency of CS | 0.15 | 0.15 |
| Number of attempts | 0.01 | 0.27 |
| Level of Puncture | 0.48 | 0.13 |
| Needle Size | 0.01 | 0.44 |
| Dosage | 0.69 | 0.04 |
| Number of CSF drops | 0.01 | 0.57 |
| Previous history of PDPH | 0.17 | 0.15 |

## Study limitations

This study was not population-based and mainly included women who underwent Caesarean Section at Edward Francis Small Teaching Hospital (EFSTH). As a result, we were not able to apply the findings of the research to the entire Gambian population. Also the study population was small, the calculated sample size was not met.

## Supporting information

**S1.** COMPILATION data on PDPH(1).
(XLSX)

## Author contributions

**Conceptualization:** Admire Coker.

**Formal analysis:** Simon Donkor.

**Methodology:** Admire Coker.

**Writing – original draft:** Matthew Anyanwu.

**Writing – review & editing:** Matthew Anyanwu.

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
