## [Decision Letter · Decision Letter 0]

30 Oct 2024

PONE-D-24-11503
Prevalence of Dural Puncture headache after caesarean section at a Tertiary Hospital in the Gambia.
PLOS ONE

Dear Dr. Anyanwu,

Thank you for submitting your manuscript to PLOS ONE. After careful consideration, we feel that it has merit but does not fully meet PLOS ONE’s publication criteria as it currently stands. Therefore, we invite you to submit a revised version of the manuscript that addresses the points raised during the review process.

We look forward to receiving your revised manuscript.

Kind regards,

Sagar Panthi, MBBS

Academic Editor

PLOS ONE

Journal Requirements:

4. Please amend the manuscript submission data (via Edit Submission) to include author Donkor Simon.

Reviewers' comments:

Reviewer's Responses to Questions

**Comments to the Author**

1. Is the manuscript technically sound, and do the data support the conclusions?

Reviewer #1: Yes

Reviewer #2: No

Reviewer #3: Yes

Reviewer #4: Yes

2. Has the statistical analysis been performed appropriately and rigorously? 

Reviewer #1: Yes

Reviewer #2: No

Reviewer #3: Yes

Reviewer #4: Yes

3. Have the authors made all data underlying the findings in their manuscript fully available?

Reviewer #1: Yes

Reviewer #2: No

Reviewer #3: No

Reviewer #4: Yes

4. Is the manuscript presented in an intelligible fashion and written in standard English?

Reviewer #1: Yes

Reviewer #2: No

Reviewer #3: Yes

Reviewer #4: Yes

5. Review Comments to the Author

Reviewer #1: Dear authors, thank you for your collaboration.

I consider it to be of interest and well developed, since it addresses the issue of headache after the use of epidural anesthesia, one of the undesirable effects in such emotional moments as the birth of a new family member.

Reviewer #2: The manuscript has many major issues:

1- The language is very poor and MAJOR edits needs to be done. I suggest getting help form an expert English editor or a native speaker before trying to re-submit.

2- The methodology is unclear. I can't tell if that was truly cross-sectional, retrospective or prospective study.

3- The sample size calculation was not reproducible using the same input reported by the authors.

4- The Introduction part has a plagiarism issue with about 60% of the text being not genuine.

5- I did not understand why are you reporting participants' religion in your data. If this has any thing to do with the outcome, it must be justified.

Good luck with your next submission

Reviewer #3: Although this study is not novel, it could add to information in obstetrics practice

ABSTRACT: The authors stated:"lasting for 3hrs (18%)". Since this is a retrospective study, how did the authors document the duration of the post dural headache.

Introduction

The authors should beef up the justification for the study.

METHODS

How did the authors ensure uniformity of patient recruitment? how was the sample size calculated? Was one singe type of spinal needle used in all the participants?

What is the ethical approval number?

Reviewer #4: i have added an attachment with comments as a separate file, kindly go through the file

few of them are: keywords to be arranged alphabetically

address the typographical errors

below 18 years are considered as minor and you have included young girls of 15 years, how do you justify it?

6. PLOS authors have the option to publish the peer review history of their article (what does this mean?). If published, this will include your full peer review and any attached files.

Reviewer #1: No

Reviewer #2: No

Reviewer #3: No

Reviewer #4: **Yes: **Gajal Lakhe

---

## [Author Response · Author response to Decision Letter 1]

24 Dec 2024

Response to the reviewers: Date: 6th December 2024

We sincerely thank all the reviewers of this article. Is our pleasure to provide responses to the comments and try as much as possible to factor all corrections in the reversed version of the main manuscripts.

Reviewer #1:

comment; I consider it to be of interest and well developed, since it addresses the issue of headache after the use of epidural anesthesia, one of the undesirable effects in such emotional moments as the birth of a new family member.

Response: We thank you for your kind and supportive comment towards the article. Thank you so very much.

Reviewer #2: The manuscript has many major issues:

1- The language is very poor and MAJOR edits needs to be done. I suggest getting help form an expert English editor or a native speaker before trying to re-submit.

Response: thank you dear reviewer will have reviewed the language with the help of an expert in the language.

2- The methodology is unclear. I can't tell if that was truly cross-sectional, retrospective or prospective study.

Response: the study was a cross sectional study. A cross sectional study can be retrospective or prospectively conducted. In our study it was prospectively conducted.

3- The sample size calculation was not reproducible using the same input reported by the authors.

Response: G power 3.1 software inputs was set at the numbers desired for the study and the output was more than the number we recruited because of the duration of the study.

4- The Introduction part has a plagiarism issue with about 60% of the text being not genuine.

Response : that has been reversed thanking you for the observations.

5- I did not understand why are you reporting participants' religion in your data. If this has any thing to do with the outcome, it must be justified

Response: we do not have justification other than a biodata that was included in the data collection tool. It has no relationship with the subject.

Reviewer #3:

Comments 1: Although this study is not novel, it could add to information in obstetrics practice

ABSTRACT: The authors stated:"lasting for 3hrs (18%)". Since this is a retrospective study, how did the authors document the duration of the post dural headache.

Response: It was a cross sectional study The data collection and observation of study participants was prospectively conducted.

Comment 2: introduction; the authors should beef up the justification for the study.

Response: thank you for the observation we have added more sentences to justify the study.

Comment 3: METHODS

How did the authors ensure uniformity of patient recruitment? how was the sample size calculated? Was one singe type of spinal needle used in all the participants?

What is the ethical approval number?

Response: patients were recruited consecutively as they were admitted in the labour or antenatal wards. The study identification number is issued when decision to undergo caesarean section was reached. The sample size was calculated using a G power 3.1 software.

Reviewer #4: i have added an attachment with comments as a separate file, kindly go through the file

few of them are: keywords to be arranged alphabetically

address the typographical errors below 18 years are considered as minor and you have included young girls of 15 years, how do you justify it?

Response:

Thank you for your comments. The keywords has been arranged accordingly, the typo errors rectified as much as possible and no study participant that was 15 years. The 15 years was a definition of the lower limit of the age range of reproductive age.

---

## [Decision Letter · Decision Letter 1]

4 Mar 2025

PONE-D-24-11503R1

Prevalence of Dural Puncture headache after caesarean section at a Tertiary Hospital in the Gambia.

PLOS ONE

Dear Dr. Anyanwu,

Thank you for submitting your manuscript to PLOS ONE. After careful consideration, we feel that it has merit but does not fully meet PLOS ONE’s publication criteria as it currently stands. Therefore, we invite you to submit a revised version of the manuscript that addresses the points raised during the review process.

A marked-up copy of your manuscript that highlights changes made to the original version. You should upload this as a separate file labeled 'Revised Manuscript with Track Changes'.An unmarked version of your revised paper without tracked changes. You should upload this as a separate file labeled 'Manuscript'.

We look forward to receiving your revised manuscript.

Kind regards,

Sagar Panthi, MBBS

Academic Editor

PLOS ONE

Journal Requirements:

Additional Editor Comments:

Dear Authors,

Thank you for submitting the revision of the manuscript and addressing most of the reviewers' comments. While the manuscript has potential merits and is eligible for being accepted into publication with Plos One, few of the comments from Reviewer 4 in previous revision have not been addressed so I would request the authors to carefully see through the attached file with the comments from Reviewer 4. In addition, the language used in the manuscript is still substandard and hence I would request to get professional english editing and re-submit the revision so that we can move forward with the processing of this manuscript.

In case of any queries, please feel free to reach out to the editorial team at Plos One.

Regards,

Sagar Panthi, MBBS

Academic Editor, Plos One

Reviewers' comments:

Reviewer's Responses to Questions

**Comments to the Author**

1. If the authors have adequately addressed your comments raised in a previous round of review and you feel that this manuscript is now acceptable for publication, you may indicate that here to bypass the “Comments to the Author” section, enter your conflict of interest statement in the “Confidential to Editor” section, and submit your "Accept" recommendation.

Reviewer #1: All comments have been addressed

Reviewer #3: All comments have been addressed

2. Is the manuscript technically sound, and do the data support the conclusions?

Reviewer #1: Yes

Reviewer #3: Yes

3. Has the statistical analysis been performed appropriately and rigorously? 

Reviewer #1: Yes

Reviewer #3: Yes

4. Have the authors made all data underlying the findings in their manuscript fully available?

Reviewer #1: Yes

Reviewer #3: Yes

5. Is the manuscript presented in an intelligible fashion and written in standard English?

Reviewer #1: Yes

Reviewer #3: Yes

6. Review Comments to the Author

Reviewer #1: Dear authors, thank you for your contribution.

This is an interesting area where you have conducted an interesting local survey.

Reviewer #3: The authors have addressed the comments raised. The authors stated that they have beefed up the introduction.

7. PLOS authors have the option to publish the peer review history of their article (what does this mean?). If published, this will include your full peer review and any attached files.

Reviewer #1: No

Reviewer #3: No

---

## [Author Response · Author response to Decision Letter 2]

5 Aug 2025

Response to the reviewers number 1-3 was well attended but this is for reviewer number 4: Date: 11th March, 2025

We sincerely thank all the reviewers of this article. Is our pleasure to provide responses to the comments and try as much as possible to factor all corrections in the second reversed version of the main manuscripts.

Reviewer #4:

Reviewer #4: i have added an attachment with comments as a separate file, kindly go through the file

few of them are: keywords to be arranged alphabetically

address the typographical errors below 18 years are considered as minor and you have included young girls of 15 years, how do you justify it?

Response:

Thank you for your comments.

The file has been downloaded.

Response: The keywords has been arranged accordingly

the typo errors rectified as much as possible and no study participant that was 15 years. The 15 years was a definition of the lower limit of the age range of reproductive age.

The rest of the comments and response has been delivered through the file.

Thank you.

Matthew Anyanwu

Corresponding author

---

## [Decision Letter · Decision Letter 2]

13 Aug 2025

Prevalence of dural puncture headache after caesarean section at a Tertiary Hospital in the Gambia.

PONE-D-24-11503R2

Dear Dr. Anyanwu,

We’re pleased to inform you that your manuscript has been judged scientifically suitable for publication and will be formally accepted for publication once it meets all outstanding technical requirements.

Kind regards,

Siraj Ahmed Ali

Academic Editor

PLOS ONE

Additional Editor Comments (optional):

Reviewers' comments:

Reviewer's Responses to Questions

**Comments to the Author**

1. If the authors have adequately addressed your comments raised in a previous round of review and you feel that this manuscript is now acceptable for publication, you may indicate that here to bypass the “Comments to the Author” section, enter your conflict of interest statement in the “Confidential to Editor” section, and submit your "Accept" recommendation.

Reviewer #1: All comments have been addressed

Reviewer #3: (No Response)

2. Is the manuscript technically sound, and do the data support the conclusions?

Reviewer #1: Yes

Reviewer #3: Yes

3. Has the statistical analysis been performed appropriately and rigorously? 

Reviewer #1: Yes

Reviewer #3: Yes

4. Have the authors made all data underlying the findings in their manuscript fully available?

Reviewer #1: Yes

Reviewer #3: Yes

5. Is the manuscript presented in an intelligible fashion and written in standard English?

Reviewer #1: Yes

Reviewer #3: No

6. Review Comments to the Author

Reviewer #1: Dear Authors, thank you for continuing to improve your paper based on the reviewers' recommendations.

I wish you success.

Reviewer #3: Overall Impression

The discussion contains important comparisons with other studies and acknowledges factors influencing PDPH prevalence (needle size, number of attempts, type of CS, sample size). However:the section is too repetitive,

Weaknesses / Issues:

Lack of thematic organization:

Needle gauge is discussed in multiple scattered places rather than consolidated into one clear sub-section.

Number of attempts is introduced twice with overlapping text.

Emergency vs elective CS is addressed abruptly without linking to needle size or operator skill level.

Excessive repetition:

The Jordan study (6.3% incidence) is mentioned twice almost verbatim.

The association between needle gauge and PDPH is described multiple times.

Weak critical interpretation:

While many external studies are cited, the discussion does not critically analyse why the prevalence in The Gambia is so much higher than in most cited studies beyond gauge and attempts.

No exploration of other possible local factors such as operator training, hydration status, BMI, or post-operative positioning.

No mention of possible biases (selection bias, recall bias for headache duration).

Limited discussion of limitations:

Small sample size is acknowledged but not fully explored in terms of its impact on confidence intervals, statistical power, and generalisability.

The potential effect of being a single-centre study is not mentioned.

Poorly structured (frequent jumping between topics without logical progression), and

Contains numerous grammatical errors and unclear phrasing that weaken clarity and impact.Language and Style Issues

Grammar: Frequent misuse of tenses (“was a similar findings” → “was a similar finding”; “dura matter” → “dura mater”).

Awkward phrasing:

“with low threshold for pain” should be scientifically supported and carefully worded to avoid assumptions or bias.

“majority of the PDPH” is unclear—should be “majority of PDPH cases.”

Run-on sentences: Many sentences are overly long, with multiple ideas strung together without proper punctuation.

Recommendations for Improvement

Restructure for clarity:

Suggested order:

Restate main finding (42.7% prevalence) and briefly summarise its significance.

Compare with other African studies (Ethiopia, Nigeria, Ghana) and discuss similarities/differences.

Compare with other international studies and analyse potential reasons for differences.

Detailed sub-section on needle gauge effect.

Detailed sub-section on number of attempts.

Discussion of CS type (emergency vs elective) and possible confounders.

Pain severity comparison.

Limitations and strengths of study.

Clinical implications and recommendations.

Eliminate duplication: Merge repeated references to the same studies into single, well-analysed points.

Improve scientific depth: Include other possible contributing factors to high PDPH prevalence (hydration, postoperative mobilization time, anaesthetist experience).

Language polishing: Shorten sentences, correct grammar, and improve transitions between ideas.

Statistical reporting: Add confidence intervals and clarify the strength of associations.

7. PLOS authors have the option to publish the peer review history of their article (what does this mean?). If published, this will include your full peer review and any attached files.

Reviewer #1: No

Reviewer #3: No

---

## [Editor Report · Acceptance letter]

PONE-D-24-11503R2

PLOS ONE

Dear Dr. Anyanwu,

I'm pleased to inform you that your manuscript has been deemed suitable for publication in PLOS ONE. Congratulations! Your manuscript is now being handed over to our production team.

Kind regards,

on behalf of

Dr. Siraj Ahmed Ali

Academic Editor

PLOS ONE